# Breeding of Buckwheat for Usage of Sprout and Pre-Harvest Sprouting Resistance

**DOI:** 10.3390/plants10050997

**Published:** 2021-05-17

**Authors:** Tatsuro Suzuki, Takahiro Hara, Kenjiro Katsu

**Affiliations:** Kyushu Okinawa Agricultural Research Center, NARO, Suya 2421, Koshi, Kumamoto 861-1192, Japan; hrtk@affrc.go.jp (T.H.); kkatsu9699@affrc.go.jp (K.K.)

**Keywords:** buckwheat, seed, sprouting, dormancy

## Abstract

Buckwheat is recognized as an important traditional crop and supports local economies in several regions around the world. Buckwheat is used, for example, as a cereal grain, noodle and bread. In addition, buckwheat is also used as a sprout or a young seedling. For these foods, sprouting is an important characteristic that affects food quality. For foods made from buckwheat flour, pre-harvest sprouting may decrease yield, which also leads to the deterioration of noodle quality. Breeding buckwheat that is resistant to pre-harvest sprouting is therefore required. Germination and subsequent growth are also important characteristics of the quality of sprouts. Although buckwheat sprouts are the focus because they contain many functional compounds, such as rutin, several problems have been noted, such as thin hypocotyls and husks remaining on sprouts. To date, several new varieties have been developed to resolve these quality issues. In this review, we summarize and introduce research on the breeding of buckwheat related to quality, sprouting and subsequent sprout growth.

## 1. Introduction

Buckwheat is an important cereal in Japan, China, Russia, Europe and elsewhere [1,2]. In some regions, buckwheat is recognized as an important traditional crop that supports local economies. Buckwheat is generally used as a grain cereal after removal of the husk (cereal grain), and is sometimes milled for flour. The flour is generally processed into other foods, such as noodles, bread and confectionery. In addition, sprouts, which are germinated young plants, are also used as vegetables. Leaves can sometimes be dried to form teas and powders.

Sprouting is an integral part of the cultivation and production process of sprouts, and is important for controlling the quality of processed foods, such as noodles and bread.

In sprouts, the germination rate and subsequent growth outcomes, including the thickness of the hypocotyl and changes in functional compounds, are important characteristics. Some sprout varieties have been developed to optimize these factors. In terms of grain use, pre-harvest sprouting is one of the most important characteristics, especially in spring-sown buckwheat, particularly in Japan. Pre-harvest sprouting sometimes occurs when maturing seeds on plants encounter substantial rain and high temperatures, which leads to a decrease in yield and deterioration in quality. Japanese buckwheat breeders have developed varieties with pre-harvest sprouting resistance [3]. The dormancy of seeds in plants is an important adaptive trait that is a primary component of different life history strategies [4]. To date, many aspects of seed dormancy and germination have been reviewed [5,6]. Understanding the mechanism of germination and dormancy is also important for breeding new varieties of buckwheat with improved sprouting-related characteristics.

In this review, we summarize some important points in terms of buckwheat breeding and introduce the current research and varieties on buckwheat quality for sprouting and discuss future prospects.

## 2. Seed Dormancy and Germination in Buckwheat

In plants, primary seed dormancy refers to the innate dormancy of seeds when they are separated from the plant. In buckwheat, the occurrence of both primary seed dormancy and secondary seed dormancy has been reported [7]. For each plant species, germination (loss of primary dormancy) varies according to the applied conditions.

In buckwheat, the degree of loss of primary dormancy mainly depends on the storage period and temperature, and a long storage period and high storage temperature are required to lose primary dormancy [7]. Varietal differences in germination rates have also been reported [8,9,10]. Hara et al. (2008) [7] reported that cultivar differences in buckwheat pre-harvest sprouting were correlated with seed dormancy based on a petri dish experiment. The petri dish assay can be used to evaluate the pre-harvest sprouting resistance of buckwheat cultivars, and it has been employed for selection in buckwheat breeding programs, even when pre-harvest sprouting does not occur under natural rainfall. In the Petri dish assay, seeds that did not germinate immediately after sampling were considered dormant, and most of the seeds germinated after 6 months of storage under chilling conditions or room temperature. These findings indicate that cultivar differences in buckwheat pre-harvest sprouting are explained mainly by seed dormancy. In buckwheat, the seed germination rate increases with temperature; for example, germination is higher at 28 °C than at 20 °C. Hara et al. (2007, 2009a) [11,12] reported the presence of resistance to pre-harvest sprouting in buckwheat cultivars, and they evaluated cultivar differences in pre-harvest sprouting in the rainy season in an experimental field by counting sprouted grains harvested after frequent natural rainfall. The ratio of the pre-harvest sprouted grains ranged from 28% to 81%. The germination rate at harvesting time in common and Tartary buckwheat was also investigated using the Petri dish test [7]. Germination rates were different for different varieties and species. Among the common buckwheat varieties, ‘Harunoibuki’ (37.4%) and ‘Hitachiakisoba’ (21.1%) had lower germination rates than the other varieties (70.0%–90.0%) at 25 °C. In addition, all the common buckwheat varieties had higher germination rates (85.5–98.9%) at 35 °C than at 25 °C. The germination of Tartary buckwheat was lower than that of common buckwheat at both 25 °C and 35 °C. The germination of ‘Latvia’, ’Hokkai T8′ and ‘Manten-Kirari’ was significantly lower than that of other Tartary buckwheat except for ‘Hokkai T9′ in Ryan’s multiple range test (*p* < 0.05). This report is the first to show varietal differences in primary dormancy in Tartary buckwheat. Germination at 25 °C in Tartary buckwheat was lower than that at 35 °C. Among the Tartary buckwheat varieties, ‘Latvia’ (43.3%) and ‘Manten-Kirari’ (40.0%) had lower germination rates than the other Tartary buckwheat varieties.

Hara et al. [11,12] investigated the effect of pre-harvest sprouting on the textural characteristics of buckwheat noodles and pasting viscosity. The sprouting caused a decrease in the peak force and peak strain when noodles made from buckwheat flour or buckwheat plus wheat flour were cut. Hara et al. (2007) [11] demonstrated that pre-harvest sprouting lowered the RVA (rapid visco analyzer) peak viscosity because it digested buckwheat starch by alpha-amylase [13]. The decrease in peak viscosity caused by sprouting is not complemented by the addition of wheat flour [11] because α-amylase in sprouted buckwheat flour can digest wheat starch [14]. Under this background, a variety with pre-harvest sprouting resistance, that is, strong primary seed dormancy, must be developed.

Secondary seed dormancy refers to a dormant state that is induced in seeds under unfavorable germination conditions [15]. In buckwheat, secondary seed dormancy has also been studied by Suzuki et al. [7], who stored buckwheat seeds harvested at 5, 25 and 35 °C for 0, 10 and 30 days respectively. The germination rate was subsequently evaluated using Petri dishes at 25 °C and 35 °C. In common buckwheat, the germination rate increased or reached a plateau during storage at 25 °C and 35 °C. When the seeds were stored at 5 °C and germination was tested at 25 °C, germination of the ‘Hitachiakisoba’ and ‘Harunoibuki’ seeds was lower at maturation compared to the other buckwheat varieties and increased after 10 days of storage, which indicates that these two varieties have stronger primary seed dormancy than the others. In Tartary buckwheat, the germination of the Chinese variety increased or reached a plateau in all treatments. ‘Hokkai T8′, ‘Manten-Kirari’, ‘Hokkai T9′ and ‘Latvia’ seeds stored at 5 °C for 10 days showed decreased germination at 25 and 35 °C. For seeds stored for 30 days, the germination of ‘Hokkai T8′, ‘Manten-Kirari’ and ‘Hokkai T9′ increased while the seeds of ‘Latvia’ stored at 5 °C still did not germinate. This finding indicates that secondary dormancy should occur in Tartary buckwheat, but not in common buckwheat. This report is the first to demonstrate secondary dormancy in buckwheat seeds, and the authors hypothesized that it may be related to the increased adaptability strategy of buckwheat species for high altitude or northern regions, where seed secondary dormancy has advantages, such as ‘frost risk escape’ in the winter season. Strong seed dormancy is sometimes not beneficial for buckwheat cultivation because short periods occur between harvest and the next sowing.

Abscisic acid (ABA) is a well-known plant hormone that plays a fundamental role in the induction and maintenance of dormancy, including the control of seed germination [16,17]. In addition, its synthesis and degradation are closely related to the control of seed dormancy and germination. In buckwheat, changes in abscisic acid are important for understanding the mechanism of pre-harvest sprouting. Recently, some important genes, including transcription factors that regulate ABA concentration, have been studied, including viviparous 1 (*Vp1*) [18,19], delay of germination 1 (*DOG1*) [20], biosynthesis (*ABA1*, *ABA2* and *NCEDs*) [21] and catabolism (*CYP707A*) [21]. In buckwheat, these points should be addressed and clarified in the future.

## 3. Breeding for Buckwheat Sprout with High Quality

Seed sprouts are an atypical vegetable stage, and they have received attention as functional vegetables because of their beneficial nutritive value associated with amino acids, fibers, minerals and proteins [22]. In the Japanese market, a variety of different types of sprouts can be found, including broccoli (*Brassica oleracea* L. var. *italica* Plenck.), common buckwheat (*Fagopyrum esculentum* Munch), kale (*B. oleracea* L. var. *encephala*), mung bean (*Phaseolus aureus* Rob.), red cabbage (*B. oleracea* L. var. *capitata f. rubra*) and soybean (*Glycine max* (L.) Merr.). Buckwheat sprouts (seedlings) are suitable for use as vegetables because they contain large amounts of functional compounds [23,24]. The phenolic composition differs between common buckwheat and Tartary buckwheat sprouts, and compared to the sprouts of common buckwheat, those of Tartary buckwheat have recently received greater attention as a functional food because of the greater content of rutin, a flavonol glycoside [25,26,27]. Kim et al. (2007) [27] investigated the phenolic compositions of nongerminated/germinated seeds and seed sprouts (6–10 days old) of common and Tartary buckwheat. In the edible parts of common buckwheat sprouts, individual phenolics significantly increased during sprout growth from 6 to 10 days after sowing (DAS), whereas in Tartary buckwheat sprouts, they did not. While the sum contents of phenolic and edible parts (mean 24.4 mg/g DW at 6–10 DAS) of Tartary buckwheat sprouts were similar to those of common buckwheat sprouts, the rutin content in the nongerminated/germinated seeds (mean 14.7 mg/g DW) and edible parts (mean 21.8 mg/g DW) of Tartary buckwheat were 49- and 5-fold higher than those of common buckwheat, respectively. Extracts of the edible parts of both species showed very similar free radical-scavenging activities (mean 1.7 lmol Trolox eq/g DW), suggesting that the overall antioxidative activity might be affected by the combination of identified phenolics and unidentified (minor) components. In addition, the effect of light on the accumulation of functional compounds has also been reported [25]. Although buckwheat sprouts are good food materials, production factories have pointed out several problems that need to be improved as follows: (1) thin hypocotyls compared to other sprouts, such as radish sprouts, (2) high ‘husk-remaining’ rates on germinated sprouts, which worsen the texture and (3) only green cotyledons, especially in Tartary buckwheat sprouts.

‘Hokkai T9′ was developed by selection from a genetic resource, ‘Dattan-shu’, treated with colchicine. In 1994, breeders treated colchicine with ‘Dattan-shu’ and obtained 127 seeds. The next year, they selected 29 tetraploid individuals by observing chloroplasts from progeny individuals. After 1996, they selected and named a promising line with a large seed size as ‘Hokkai T9′. ‘Hokkai T9′ has thick hypocotyl compared to diploid varieties such as ‘Hokkai T8′ and ‘Hokkai T10′ (Figure 1). ’Hokkai T10′ was developed from a mutant of ‘Hokkai T8′ treated with ethyl methane sulfonate (EMS). In 2001, we treated EMS with ‘Hokkai T8′ and obtained approximately 2600 seeds. In the M3 generation, we found and selected one individual that had a red cotyledon and hypocotyl in the seedling growth stage. After 2002, we selected and developed a variety with red cotyledon and hypocotyl traits (Figure 1) up to the M10 generation. The rutin concentration in ‘Hokkai T9′ and T10” was higher than that in “Hokkai T8”. In addition, the anthocyanin concentration of ‘Hokkai T10′ was far higher than that of the other varieties. This trait is dominated by a single recessive gene [28]. ‘Hokkai T9′ is a tetraploid variety; therefore, the hypocotyl diameter and folds would be higher than those of the other varieties, indicating that the sprout of ‘Hokkai T9′ was promising as a raw sprout in terms of appearance. In addition, the remaining husk percentage of ‘Hokkai T9′ was only 2.2%, whereas that of ‘Hokkai T8′ and T10 was approximately 10%. In Japan, the husk remaining on raw sprouts is a major problem. Therefore, the tetraploid variety was suitable for raw sprouting. In addition, it has been reported that the accumulation pattern of some metal ions, such as cadmium, differs from that of diploid and tetraploid buckwheat [29]. Cadmium uptake by the aerial parts of buckwheat was insignificant and was not influenced by the ploidity level, whereas the root systems of tetraploid plants demonstrated an increased ability to accumulate cadmium compared to the roots of diploid buckwheat. This information should be useful for controlling the risk of sprout intake. Moreover, the rice Tartary buckwheat type could be useful because it can be easily dehulled by rubbing the seeds between the fingers [30]. ‘Hokkai T10′ has a red cotyledon and hypocotyl. In addition, lyophilized powder also had a red color. The sprouts of ‘Hokkai T10′ may thus be useful as anthocyanin-rich foodstuffs. In recent years, Tartary buckwheat, such as seed (green) sprouts, has been commercialized in goods sold in Japan, and the lyophilized powder has been used in alcohol, juice and teas rich in rutin. Lyophilized powder is widely mixed with other foods, including bread or confectioneries. The lyophilized powder of ‘Hokkai T10′ would constitute a high-quality additive, given its deep red color. In buckwheat, anthocyanins can be found in all aboveground organs, including cotyledons and leaves [31]. Variations in anthocyanin content according to plant organs and growth stages of buckwheat would have important implications for human health.

In Japan, most ‘Moyashi’, a generic name of sprouts grown in darkness, are traditional foods with soft textures. However, ‘Moyashi’ only has a yellowish color; therefore, other colored sprouts, even those grown in darkness, are required. Suzuki et al. (ISBS2^nd^) investigated the effect of light on the growth of ‘Hokkai T10′, ‘Hokkai T8′ and ‘Hokkai T10′ and anthocyanin accumulation. Nine DAS light-grown ‘Hokkai T10′ sprouts accumulated 0.13 and 4.92 mg/g dry wt cyanidin 3-*O*-glucoside and cyanidin 3-*O*-rutinoside, respectively. Nonchlorophyll-accumulating dark-grown sprouts accumulated 0.09 and 2.77 mg/g dry wt cyanidin 3-*O*-glucoside and cyanidin 3-*O*-rutinoside, whereas ‘Hokkai T8′ accumulated only trace amounts of anthocyanins. In general, most dark-grown plants accumulate fewer anthocyanins than light-grown plants. Common and Tartary buckwheat varieties/breeding lines accumulated only trace amounts of anthocyanins, whereas dark-grown ‘Hokkai T10′ accumulated 56% of the cyanidin 3-*O*-rutinoside compared to light-grown conditions, which indicates that ‘Hokkai T10′ may have a unique regulation mechanism for dark accumulation of anthocyanins compared with other plants since the hypocotyls and cotyledons of ‘Hokkai T10′ are deep red, even in darkness. Therefore, ‘Hokkai T10′ would be promising as colored sprouts, including ‘Moyashi’. Sprouts of ‘Hokkai T9′ and ‘Hokkai T10′ have many advantages, such as high rutin concentration, good appearance and unique red color.

On the other hand, many studies have reported the effects of substances on the quality of sprouts in buckwheat. Debski et al. (2021a) [32] reported the effects of sodium silicate on the fatty acid composition in buckwheat sprouts. Buckwheat sprouts were soaked daily for six days in an aqueous solution of the elicitors. After seven days of sprouting, the fatty acid composition was analyzed. The results showed significant changes in the composition of fatty acids in sprouts compared to buckwheat seeds. The myristic (C14:0), palmitic (C16:0), stearic (C18:0) and behenic (C20:0) acids in sprouts increased along with the growth stages of sprouts, however the use of elicitors inhibited this tendency. Debski et al. (2021b) [33] reported the effects of sodium silicate and iron chelate on the accumulation of phenolic compounds and minerals in buckwheat sprouts. Among the major flavonoids in the buckwheat sprouts treated with Fe-EDTA, the content of esters of (-)-epicatechin and glycosides of quercetin and free forms, and the total content of flavonoids, decreased. Elicitation of buckwheat sprouts with EDTA and Fe-EDTA reduced the contents of Zn, Ca, Cu, and K. Fe-EDTA treatment caused a 200% increase in the silicon content and a 500% increase in the iron content. Conversely, elicitation with both sucrose and CaCl2 in buckwheat sprouts enhanced the accumulation of bioactive compounds, such as polyphenols, flavonoids, gamma-aminobutyric acid and vitamin C and E, without negatively affecting sprout growth [34]. In addition, elicitation with both sucrose and CaCl2 significantly enhanced the antioxidant activities. Slightly acidic electrolyzed water (SAEW) also influences the accumulation of functional compounds in buckwheat [35]. The buckwheat treated with a low available chlorine concentration of 10.94 mg/L showed the best accumulation of the bioactive substance; 63.0%, 113.07% and 128.1% higher total phenolics, total flavonoids, and rutin content on the 9th day, respectively, compared to the control. These findings suggest that elicitation with external substances can produce buckwheat sprouts with a high content of flavonoids and iron. Kim et al. (2020) [36] demonstrated that the phenylpropanoid biosynthetic pathway is stimulated under abiotic stress, resulting in a higher accumulation of various phenylpropanoid compounds. In addition, light sources, including laser light, can also change functional compounds in buckwheat sprouts [37,38]; blue light significantly enhanced the contents of C-glycosylflavones such as orientin, vitexin and their isomers, and rutin. Sprouts grown under blue light also exhibited the highest total phenolic and flavonoid contents. Isoorientin is the highest antioxidant flavonoid, although numerous studies have suggested that rutin is a typical antioxidant compound in common buckwheat. These results demonstrated that blue light could be applied to enhance not only antioxidant activity but also flavonoid content in common buckwheat sprouts [37]. When common buckwheat and Tartary buckwheat sprouts were treated with a laser (He-Ne laser, 632 nm, 5 mW), out of 49 targeted minerals, vitamins, pigments and antioxidants, more than 35 parameters were significantly increased in these sprouts. In addition, laser light enhanced antioxidant capacity and anti-inflammatory activities, particularly in Tartary buckwheat sprouts [38]. Accordingly, light sources, including laser light, may be promising for improving the nutritional and health-promoting properties of buckwheat sprouts. Witkowicz et al. (2020) [39] analyzed the influence of plant growth promoters and biological control agents on the chemical composition and antioxidant activity of buckwheat sprouts. The highest antioxidant activity was found in the sprouts grown from seeds soaked in *Ecklonia maxima* extract and *Pythium oligandrum*. Based on these findings, buckwheat sprouts production can be optimized not only through selective breeding, but also through the creation of certain cultivation conditions. Recently, a high-efficiency emasculation method for Tartary buckwheat using hot water has been developed [40]. Therefore, we can develop new Tartary buckwheat varieties efficiently by hybridization between varieties/genetic resources.

## 4. Effects of Pre-Harvest Sprouting on Quality and Breeding for Pre-Harvest Sprouting Resistance

In Japan, buckwheat can be sown twice a year, such as before the early frosts in late summer (from August to September) and after the late frosts in spring (from March to May). Spring-sown buckwheat is of increasing importance due to its avoidance of the typhoon season (the peak typhoon season in Japan is mainly from July to November), which can lead to lodging. A significant problem with spring-sown buckwheat is pre-harvest sprouting (Figure 2) because the harvesting of spring-sown buckwheat is sometimes delayed during the rainy season (the peak rainy season in Japan is from June to July). This problem has been reported in Japan [11], Korea [41] and Australia [42]. Pre-harvest sprouting not only reduces the yield, but also the quality of buckwheat flour and processed foods [11]. For example, pre-harvest sprouting decreases pasting viscosity in buckwheat dough [11,41]. In addition, sprouted pre-harvest buckwheat grains have little market value [42]. Therefore, breeding varieties resistant to pre-harvest sprouting is required.

The mechanism of pre-harvest sprouting in buckwheat has been previously studied [7,9]. They showed varietal differences and effects of temperature on the germination rate based on primary seed dormancy. In 2008, the common buckwheat variety Harunoibuki (mass selected from the cultivar Hashikamiwase) was registered. This variety has improved pre-harvest sprouting resistance, which is suitable for summer cultivation and has the highest yield among cultivars grown at the same time. To our knowledge, this is the first trial to improve pre-harvest sprouting tolerance in buckwheat. We hypothesized that the success of this breeding effort is partly due to intravarietal diversity in pre-harvest sprouting. In 2018, a new cultivar, ‘NARO-FE-1′, which has lower pre-harvest sprouting than ‘Harunoibuki’, was developed by mass selection from the progeny of crosses between eight cultivars. They demonstrated that pre-harvest sprouting, which leads to yield reductions similar to quality deterioration, can be reduced by breeding efforts by exploiting intra- and intervarietal diversity in seed dormancy in buckwheat. In addition, genetic analysis of pre-harvest sprouting has revealed many QTLs and some causal genes in other crops, such as wheat, barley, maize and rice. These advances contribute to breeding efforts through marker-assisted selection and the pyramidization of QTLs and genes related to several different traits. In buckwheat, ‘Kyukei 28′ and ‘Kyukei 29′ have been reported as promising breeding lines with pre-harvest sprouting resistance. The major tolerance genes against pre-harvest sprouting in ‘Kyukei 28′ are recessive, and those in ‘Kyukei 29′ are dominant [43], which would help to accelerate the development of DNA markers for the selection of tolerant lines. Diversity in pre-harvest sprouting levels should be further explored to develop cultivars with even lower pre-harvest sprouting. In double cropping, summer ecotype cultivars showed a greater yield than autumn ecotype cultivars. Cultivated species are under automatic selection pressure and lose dormancy [44]. Continuous double cropping using summer ecotype cultivars may hasten the selection of lower seed dormancy because the length of time between harvesting and the next sowing is shorter than that in single cropping.

## 5. Present Problems and Future Aspects

Although ‘Harunoibuki’ and ‘NARO-FE-1′ are resistant to pre-harvest sprouting, we observed that these cultivars at the NARO experimental field in Kyushu after harvesting had pre-harvest sprouting. From these studies, we concluded that further improvements in pre-harvest sprouting resistance in buckwheat is necessary. Rainfall at seeding time and/or harvesting time can sometimes cause delays in these procedures. To consider such situations, we further investigated the average temperature between late May (May 20) and early July (July 10) of the last 130 years in Kumamoto City. We found zero applicable days with average temperatures of over 27 °C with more than 20 mm rainfall per day for continuous two days. In addition, the Japan Meteorological Agency has predicted that the average temperature in Japan will increase by approximately 4.5 °C by the end of the 21st century as a result of global warming. In our breeding program, pre-harvest sprouting-resistance was evaluated based on a petri dish test at 25 °C for some varieties like ‘Harunoibuki’, ‘NARO-FE-1′, ‘Kyukei 28′ and ‘Kyukei 29′. Therefore, to develop pre-harvest sprouting-resistant varieties that are suitable for such climates, resistance to 32 °C germination (27 °C plus 5 °C) is necessary. Recently, new buckwheat breeding line ‘IHK1′ has developed, which has resistance to 32 °C germination by the petri dish test. In addition, strong seed dormancy in wild buckwheat from China [8,9] and the ‘Russia homostyle’ [7] have also been reported. In the ‘Russia homostyle’, germination did not occur at 35 °C by the petri dish test, indicating that this variety would be promising to resolve pre-harvest sprouting problems in Japan. On the other hand, the wild buckwheat from China and ‘Russia homostyle’ are self-pollinating buckwheat; they have lower yields than outcrossing buckwheat because of inbreeding depression. In addition, they have seed shattering trait [8]. Therefore, buckwheat from China and ‘Russia homostyle’ should be crossed to high yield outcrossing lines and the resulting crosses would be the base populations for performing a breeding program to obtain varieties highly productive with resistance to pre-harvest sprouting. For buckwheat sprouts, farmers pointed out the following issues to be improved; thin hypocotyls, a low-dehulled rate and a variety of colors. We can develop variety for sprout with ‘easily dehulled’ and ‘red cotyledons and hypocotyls’ by crossing between ‘Hokkai T10′ with “rice-Tartary buckwheat”. Further studies on the effects of Tartary buckwheat sprout uptake on human health, including safety, are also required.

## Figures and Tables

**Figure 1 plants-10-00997-f001:**
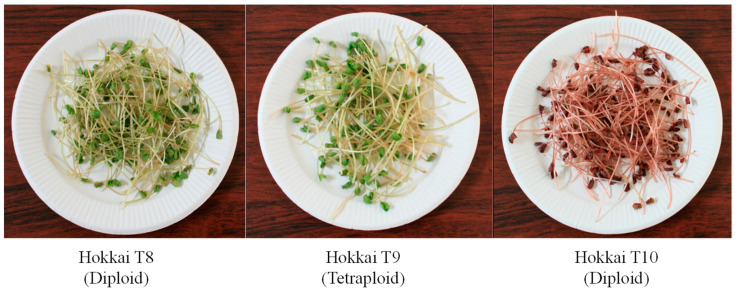
Picture of Tartary buckwheat sprouts.

**Figure 2 plants-10-00997-f002:**
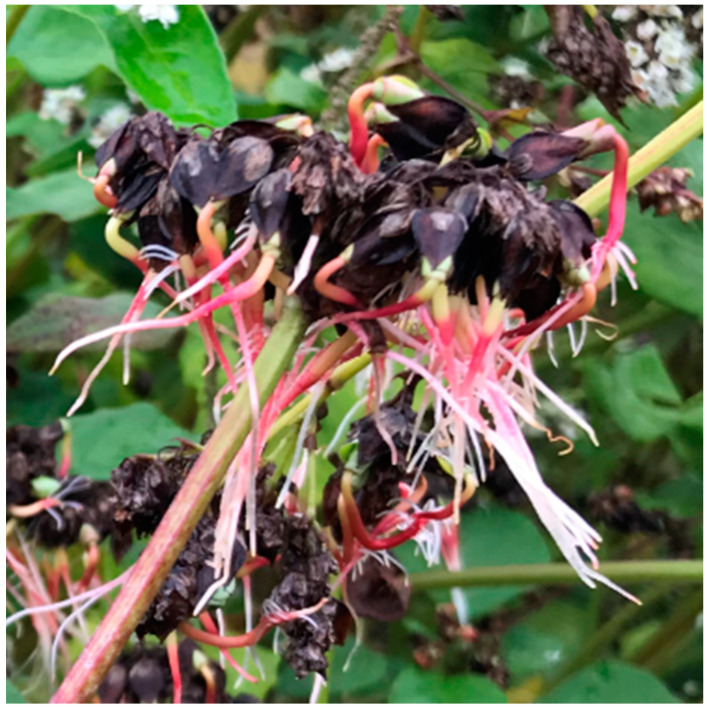
Picture of buckwheat with pre-harvest sprouting in the spring-sown buckwheat in experimental field of Kumamoto prefecture in Japan.

## Data Availability

The data that support the findings of this study are available on request from the corresponding author. The data are not publicly available due to privacy or ethical restrictions.

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
