# Peer review of "Breeding of Buckwheat for Usage of Sprout and Pre-Harvest Sprouting Resistance"

_plants, 2021, doi:10.3390/plants10050997_

Round 1

Reviewer 1 Report

Buckwheat is considered as an important traditional crop, which contributes to the development of economics in many poor regions globally. Preharvest sprouting, seed dormancy, germination rate and subsequent growth affect the buckwheat quality and functional compounds like rutin. Therefore, it is very important to develop new varieties for resolving the above problems in buckwheat. This manuscript discusses the quality of buckwheat associated with sprouting. In general, this manuscript is some simple in content and structure. Major modification should be made before consideration for publication in this journal. My concerns are listed as follows.

  1. For a review paper, a part for problems existed in update studies and future aspects regarding the topic should be included, in which authors need to put forward their opinions.
  2. A few good figures and tables should be contained in this manuscript so that audience can clearly understand the progress of buckwheat breeding on sprouting tolerance.
  3. There are only 35 references in this manuscript, among which most references were very old, published ten more years ago. Author should read and cite the latest published papers.
  4. The paper title is not attracted and confused to be understood, and needs to be revised for clear and exact meaning.
  5. There are many mistakes in grammar, spelling and format throughout the manuscript. Thereby, the whole text needs to be dramatically improved.
  6. In lines 154 and 156, the meaning for Dattan-shu and C2 should be defined. Additionally, can tetraploid individuals be determined by observing chloroplasts?
  7. The dot at the end of the paper title and “3. Breeding for buckwheat sprout with high quality” should be deleted”.

Author Response

Response to Reviewer 1 Comments

Point 1: Buckwheat is considered as an important traditional crop, which contributes to the development of economics in many poor regions globally. Preharvest sprouting, seed dormancy, germination rate and subsequent growth affect the buckwheat quality and functional compounds like rutin. Therefore, it is very important to develop new varieties for resolving the above problems in buckwheat. This manuscript discusses the quality of buckwheat associated with sprouting. In general, this manuscript is some simple in content and structure. Major modification should be made before consideration for publication in this journal. My concerns are listed as follows.

Response 1: Thank you very much for the thoughtful and constructive feedback you provided regarding our manuscript. We also appreciate the time and effort you and each of the reviewers have dedicated to providing insightful feedback on ways to strengthen our paper. Thus, it is with great pleasure that we resubmit our article for further consideration. We agree with you and revised our paper according to your suggestion.

Point 2: For a review paper, a part for problems existed in update studies and future aspects regarding the topic should be included, in which authors need to put forward their opinions.

Response 2: We emphasize them in the papers.

Point 3: A few good figures and tables should be contained in this manuscript so that audience can clearly understand the progress of buckwheat breeding on sprouting tolerance.

Response 2: We add figure 1 and Figure 2 to help understanding about traits of buckwheat variety for sprouts and preharvest-sprouting.

Point 4: There are only 35 references in this manuscript, among which most references were very old, published ten more years ago. Author should read and cite the latest published papers.

Response 2: We add nine latest references.

Point 5: The paper title is not attracted and confused to be understood, and needs to be revised for clear and exact meaning.

Response 2: We revised the title as follow; Breeding of buckwheat for usage of sprout and preharvest sprouting resistance.

Point 6: There are many mistakes in grammar, spelling and format throughout the manuscript. Thereby, the whole text needs to be dramatically improved.

Response 2: Our manuscript has edited by ‘Willy English Editing Services’.

Point 7: In lines 154 and 156, the meaning for Dattan-shu and C2 should be defined. Additionally, can tetraploid individuals be determined by observing chloroplasts?

Response 2: We add description about that ‘Dattan-shu’ is a genetic resource. Breeders confirmed that Hokkai T9 is tetraploid by observing chloroplasts.

Point 8: The dot at the end of the paper title and “3. Breeding for buckwheat sprout with high quality” should be deleted”.

Response 2: We deleted the dot.

Reviewer 2 Report

The review is quite interesting, but in my opinion it focuses mainly on the questions concerning the sprouts harvested in Japan, while some other studies may be also taken into account. There is a number of articles concerning the impact of different biostimulants and microorganisms on quality of buckwheat sprouts, thus I would suggest to add additional paragraph describing the phenomenon. Also, English should be corrected throughout the whole manuscript.

Author Response

Response to Reviewer 2 Comments

Thank you very much for the thoughtful and constructive feedback you provided regarding our manuscript. We also appreciate the time and effort you have dedicated to providing insightful feedback on ways to strengthen our paper. Thus, it is with great pleasure that we resubmit our article for further consideration. We agree with you and revised our paper according to your suggestion as follow.

Point 1: The review is quite interesting, but in my opinion it focuses mainly on the questions concerning the sprouts harvested in Japan, while some other studies may be also taken into account. There is a number of articles concerning the impact of different biostimulants and microorganisms on quality of buckwheat sprouts, thus I would suggest to add additional paragraph describing the phenomenon. Also, English should be corrected throughout the whole manuscript.

Response 1: We add description about the impact of different biostimulants and microorganisms. We also add effects of substances and light on nutritional quality of buckwheat sprouts as follow;

(Our manuscript has edited by ‘Willy English Editing Services’. )

On the other hand, many studies have reported the effects of substances on the quality of sprouts in buckwheat. Debski et al. (2021a) [32] reported the effects of sodium silicate on the fatty acid composition in buckwheat sprouts. Buckwheat sprouts were soaked daily for six days in an aqueous solution of the elicitors. After seven days of sprouting, the fatty acid composition was analyzed. The results showed significant changes in the composition of fatty acids in sprouts compared to buckwheat seeds. The myristic (C14:0), palmitic (C16:0), stearic (C18:0), and behenic (C20:0) acids in sprouts increased along with the growth stages of sprouts, however the use of elicitors inhibited this tendency. Debski et al. (2021b) [33] reported the effects of sodium silicate and iron chelate on the accumulation of phenolic compounds and minerals in buckwheat sprouts. Among the major flavonoids in the buckwheat sprouts treated with Fe-EDTA, the content of esters of (-)-epicatechin and glycosides of quercetin and free forms, as well as the total content of flavonoids, decreased. Elicitation of buckwheat sprouts with EDTA and Fe-EDTA reduced the contents of Zn, Ca, Cu, and K. Fe-EDTA treatment caused a 200% increase in the silicon content and a 500% increase in the iron content. Conversely, elicitation with both sucrose and CaCl2 in buckwheat sprouts enhanced the accumulation of bioactive compounds, such as polyphenols, flavonoids, gamma-aminobutyric acid, vitamin C, and E, without negatively affecting sprout growth [34]. In addition, elicitation with both sucrose and CaCl2 significantly enhanced the antioxidant activities. Slightly acidic electrolyzed water (SAEW) also influences the accumulation of functional compounds in buckwheat [35]. The buckwheat treated with a low available chlorine concentration of 10.94 mg/L showed the best accumulation of the bioactive substance; 63.0%, 113.07%, and 128.1% higher total phenolics, total flavonoids, and rutin content on the 9th day, respectively, compared to the control. These findings suggest that elicitation with external substances can produce buckwheat sprouts with a high content of flavonoids and iron. Kim et al. (2020)[36] demonstrated that the phenylpropanoid biosynthetic pathway is stimulated under abiotic stress, resulting in a higher accumulation of various phenylpropanoid compounds. In addition, light sources, including laser light, can also change functional compounds in buckwheat sprouts [37, 38]; blue light significantly enhanced the contents of C-glycosylflavones such as orientin, vitexin and their isomers, and rutin. Sprouts grown under blue light also exhibited the highest total phenolic and flavonoid contents. Isoorientin is the highest antioxidant flavonoid, although numerous studies have suggested that rutin is a typical antioxidant compound in common buckwheat. These results demonstrated that blue light could be applied to enhance not only antioxidant activity but also flavonoid content in common buckwheat sprouts [37]. When common buckwheat and Tartary buckwheat sprouts were treated with a laser (He-Ne laser, 632 nm, 5 mW), out of 49 targeted minerals, vitamins, pigments and antioxidants, more than 35 parameters were significantly increased in these sprouts. In addition, laser light enhanced antioxidant capacity and anti-inflammatory activities, particularly in Tartary buckwheat sprouts [38]. Accordingly, light sources, including laser light, may be promising for improving the nutritional and health-promoting properties of buckwheat sprouts. Witkowicz et al. (2020) [39] analyzed the influence of plant growth promoters and biological control agents on the chemical composition and antioxidant activity of buckwheat sprouts. The highest antioxidant activity was found in the sprouts grown from seeds soaked in Ecklonia maximaextract and Pythium oligandrum. Based on these findings, buckwheat sprouts production can be optimized not only through selective breeding, but also through the creation of certain cultivation conditions.

Round 2

Reviewer 1 Report

Authors have modified their manuscript by most comments of mine. However, authors did not well revise the content by my second suggestion "For a review paper, a part for problems existed in update studies and future aspects regarding the topic should be included, in which authors need to put forward their opinions". Authors need to add a section with a subhead like "Present problems and future aspects", in which the opinions of author on this topic should be included. In addition, the format for all the references should be kept the same including the reference titles. The dot at the end of "3. Breeding for buckwheat sprout with high quality."  should be deleted.

Author Response

Response to Reviewer 1 Comments

Thank you very much for the constructive feedback you provided regarding our manuscript. We appreciate the time and effort you have dedicated to providing insightful feedback on ways to strengthen our paper again. Tt is with great pleasure that we resubmit our article for further consideration. We agree with you and revised our paper according to your suggestion.

Point 1: Authors have modified their manuscript by most comments of mine. However, authors did not well revise the content by my second suggestion "For a review paper, a part for problems existed in update studies and future aspects regarding the topic should be included, in which authors need to put forward their opinions". Authors need to add a section with a subhead like "Present problems and future aspects", in which the opinions of author on this topic should be included.

Response 1: We add ‘5 Present problems and future aspects’ as follow;

We observed that ‘Harunoibuki’ at the NARO experimental field in Kyushu after harvesting had pre-harvest sprouting. From these studies, we concluded that further improvements in pre-harvest sprouting resistance in buckwheat is necessary. Rainfall at seeding time and/or harvesting time can sometimes cause delays in these procedures. To consider such situations, we further investigated the average temperature between late May (May 20) and early July (July 10) of the last 130 years in Kumamoto City. We found zero applicable days with average temperatures of over 27 °C. The Japan Meteorological Agency has predicted that the average temperature in Japan will increase by approximately 4.5 °C by the end of the 21st century as a result of global warming. Our breeding program, pre-harvest sprouting-resistance had evaluated based on petri dish tests at 25 °C such as for ‘Harunoibuki’, ‘NARO-FE-1’, ‘Kyukei 28’ and ‘Kyukei 29’.  Therefore, to develop pre-harvest sprouting-resistant varieties that are suitable for such climates, resistance to 32 °C germination (27 °C plus 5 °C) is necessary. In addition, strong seed dormancy in wild buckwheat from China [8,9] and the ‘Russia homostyle’ [7] have also been reported. In the ‘Russia homostyle’, germination did not occur at 35 °C by the petri dish test, indicating that this variety would be promising to resolve pre-harvest sprouting problems in Japan. For buckwheat sprouts, farmers pointed out following issues to be improved; thin hypocotyls, low-dehulled rate and variety of colors. We can develop variety for sprout with ‘easily de-hulled’ and ‘red cotyledons and hypocotyls’ by crossing between ‘Hokkai T10’ with “rice-Tartary buckwheat”. Further studies on the effects of Tartary buckwheat sprout uptake on human health, including safety, are also required.

Point 2: In addition, the format for all the references should be kept the same including the reference titles.

Response 2: We revised all the references. Some capitalized words and some italicized titles are revised.

Point 3: The dot at the end of "3. Breeding for buckwheat sprout with high quality."  should be deleted.

Response 2: We deleted the dot at the end of "3. Breeding for buckwheat sprout with high quality"

Reviewer 2 Report

The paper has been improved and now I can strongly recommend the manuscript to be published.

Author Response

Response to Reviewer 2 Comments

Thank you very much for the thoughtful and constructive feedback you provided regarding our manuscript. We also appreciate the time and effort you have dedicated to providing insightful feedback on ways to strengthen our paper. Thus, it is with great pleasure that we resubmit our article for further consideration. We agree with you and revised our paper according to your suggestion as follow.

Point 1: The paper has been improved and now I can strongly recommend the manuscript to be published.

Response 1: Thank you very much for the thoughtful comment. We also appreciate the time and effort you have dedicated to providing insightful feedback on ways to strengthen our paper.

Round 3

Reviewer 1 Report

The newly added section "5. Present problems and future aspects" is some simple and to be expanded with more opinions. The description on the new sentences need to be polished in language. For an example, "Our breeding program, pre-harvest sprouting-resistance had evaluated based on petri dish tests at 25 °C such as for ‘Harunoibuki’, ‘NARO-FE-1’, ‘Kyukei 28’ and ‘Kyukei 29’" did not make sense, should be changed to ""In our breeding program, pre-harvest sprouting-resistance was evaluated based on a petri dish test at 25 °C for some varieties like Harunoibuki, NARO-FE-1, kyukei 28 and Kyukei 29".

Author Response

Response to Reviewer 1 Comments

Thank you very much for the constructive feedback you provided regarding our manuscript. We appreciate the time and effort you have dedicated to providing insightful feedback on ways to strengthen our paper again and again. Tt is with great pleasure that we resubmit our article for further consideration. We agree with you and revised our paper according to your suggestion.

Point 1: The newly added section "5. Present problems and future aspects" is some simple and to be expanded with more opinions.

Response 1: We add following sentence.

Line 335: The wild buckwheat from China and ‘Russia homostyle’are self-pollinating buckwheat. Therefore, they have lower yields than outcrossing buckwheat because of inbreeding depression. In addition, they have seed shattering trait [8]. Development by cross breeding the wild buckwheat from China and ‘Russia homostyle’between a high yield outcrossing line is required.

Point 2: The description on the new sentences need to be polished in language. For an example, "Our breeding program, pre-harvest sprouting-resistance had evaluated based on petri dish tests at 25 °C such as for ‘Harunoibuki’, ‘NARO-FE-1’, ‘Kyukei 28’ and ‘Kyukei 29’" did not make sense, should be changed to ""In our breeding program, pre-harvest sprouting-resistance was evaluated based on a petri dish test at 25 °C for some varieties like Harunoibuki, NARO-FE-1, kyukei 28 and Kyukei 29".

Response 2: We revised the sentence according to your suggestion.

Line 325:  In our breeding program, pre-harvest sprouting-resistance was evaluated based on a petri dish test at 25 °C for some varieties like ‘Harunoibuki’, ‘NARO-FE-1’, ‘kyukei 28’ and ‘Kyukei 29’.
